# THE PATH-DRIVEN INDEPENDENCE TESTING (PIT) ALGORITHM

## ABSTRACT

PC is an efficient constraint-based algorithm for learning the structure of a Bayesian network. However, the required number of conditional independent (CI) tests can make the algorithm practically infeasible or slow for large graphs. We developed a constrained-based algorithm, called *Path-Driven Independence Testing (PIT) Algorithm*, which during the learning process, utilizes the information of the partially learned network to reduce the number of CI tests. The idea is that for each pair of variables $X$ and $Y$, instead of checking independence conditioned on every subset of all the neighbors of $X$ (resp. $Y$) as in PC, the search is restricted to only the common neighbors of $X$ and $Y$ and to neighbors connected to $Y$ (resp. $X$) by a path. Also, paths connecting $X$ and $Y$ without a descendant of a common neighbor can be blocked by observing two consecutive nodes on the path. Compared to PC, PIT is proven to conduct at most the same number of CI tests, and experimentally shown to be significantly (up to 7 times) faster and more accurate.

## 1 INTRODUCTION

Causal discovery, the task of identifying the relationship between random variables, is the key to understanding the underlying governing mechanisms and is necessary to answer interventional queries (Kitson et al., 2021). Causal relationships can be represented by a directed acyclic graph (DAG) with random variables as nodes and where links imply causal relations. Similarly, yet in a non-causal context, a DAG (the structure) together with the probability distributions of each variable conditioned on its parents (the parameters) form a *Bayesian network* over the variables, which factorizes the joint probability distribution of the variables as a multiplication of the conditional probability distributions, and in turn allows answering probabilistic queries. In either context, the task of identifying the "true" DAG from data collected on the variables is referred to as *structure learning* (Guo et al., 2020).

One approach to fulfill this task is *constraint based*, which is based on detecting (in)dependencies between the variables by performing conditional independence (CI) tests (Koller & Friedman, 2009). The idea is that two variables are not connected in the true DAG should they be independent conditioned on a subset of the other variables. The result is a class of independence-equivalence (I-equivalence) graphs that is presented as a partially DAG (PDAG). Under the Markov and faithfulness assumptions, constraint-based methods asymptotically output the "true" PDAG (Ng et al., 2021).

The PC algorithm is one of the most fundamental constraint-based approaches for causal discovery, following a two-step process (Spirtes et al., 2000). First, it identifies the direct dependencies between every pair of variables by starting from a fully connected graph and iteratively removing edges where conditional independence is detected. Second, the surviving edges are oriented to reflect causal directions. Despite its effectiveness, the number of conditional independence (CI) tests required by the PC algorithm can scale exponentially, with a worst-case complexity of $\mathcal{O}(2^N)$, where $N$ is the number of variables (Spirtes et al., 2000).

In practice, this number is often lower due to the removal of edges and reduction of adjacency sets, particularly in sparse graphs (Spirtes et al., 2000; Peters et al., 2017). However, the high number of CI tests remains a critical bottleneck (van den Boom et al., 2022; Wadhwa & Dong, 2021), especially for large-scale problems. To address this, several techniques have been developed, including distributed and parallel learning approaches (Hwang et al., 2006; Gu & Zhou, 2020; Bouhamed et al., 2015; Zarebavani et al., 2019; Le et al., 2016; Shahbazinia et al., 2023), limiting the size of conditioning sets (Sondhi & Shojaie, 2019), and pre-processing strategies (Cai et al., 2022). While these methods offer

improvements, they either demand extensive computation resources, rely on specific assumptions, or did not consistently outperform PC.

In this paper, we developed the Path-Driven Independence Testing (PDIT) Algorithm to reduce the number of CI tests in the PC algorithm. In PC, the candidate conditioning set for a pair of adjacent variables $X$ and $Y$ is every subset of the neighbors of $X$ and every subset of the neighbors of $Y$. However, some neighbors of $X$ are not connected to $Y$ and vice versa, and hence, do not render $X$ and $Y$ independent. Thus, they can be excluded from the conditioning set. On the other hand, some paths between $X$ and $Y$ may always be blocked without activating any other connecting path. Thus, these paths or a sufficient number of their nodes can be always included in the conditioning set and do not need to be searched through. PIT and its variants leverage these properties to reduce the number of candidate conditioning sets. PIT is sound and complete and never requires more CI tests than those in PC–should an oracle be used for the CI tests. In practice, our extensive evaluation across 16 datasets shows that PIT requires significantly fewer CI tests, and is significantly faster and often more accurate than PC and PC-stable (Colombo & Maathuis, 2014) as well as Hill Climbing and Tabu.

## 2 BACKGROUND

Given random variables $\mathcal{X} = \{X_1, \ldots, X_N\}$, their joint probability distribution $P(\mathcal{X})$ can be factorized according to the chain rule into $\Pi_{i=1}^{N} P(X_i \mid X_1, \ldots, X_{i-1})$. Each conditional probability term may be simplified if a corresponding conditional probability independence holds. For example, $P(X_1 \mid X_2, X_3)$ becomes $P(X_1 \mid X_2)$ should $X_1 \perp X_3 \mid X_2$ hold. The problem is to find (one of) the "simplest" factorization(s) of the joint probability distribution.

A directed acyclic graph (DAG) $\mathcal{G}$ can be attributed to each factorization, where nodes are the random variables $\mathcal{X}$ and for each conditional term $P(X_i \mid X_{i_1}, \ldots, X_{i_k}), k \geq 1$, there is an incoming link from each of the conditioned variables $X_{i_1}, \ldots, X_{i_k}$ to $X_i$. Thus, $X_{i_1}, \ldots, X_{i_k}$ form the parents of $X_i$, denoted by the set $\mathrm{Pa}_{X_i}$. A sequence of nodes $\mathcal{T} = (X_1, X_2, \ldots, X_n), n \geq 1$, where $X_i$ and $X_{i+1}$ are linked in $\mathcal{G}$ for $i = 1, \ldots, n-1$, is referred to as a *trail (path) between $X_1$ and $X_n$*. The length of the trail is the number of its links, i.e., $n-1$. Define the *interior* of the trail as $\mathrm{int}(\mathcal{T}) = \{X_2, \ldots, X_{n-1}\}$, that is, the set of all but the ending nodes. The *descendants* of a node $X$ are those nodes to which $X$ is connected by a directed path, including $X$ itself. The *reachable set of node $Y$*, denoted $\mathcal{R}_Y$, is the set of nodes, including $Y$, that are connected to $Y$ by some trail. Path $\mathcal{T}$ is a *directed path from $X_1$ to $X_n$* if node $X_i$ is linked to node $X_{i+1}$ for all $i = 1, \ldots, n-1$. Define a *collider* as a length-two trail $(X, Z, Y)$ where nodes $X$ and $Y$ are linked to $Z$, i.e., $X \to Z \leftarrow Y$. Node $Z$ is referred to as the *collider node* or *center*. The collider is an *immorality* if there is no edge between $X$ and $Y$, referred to as a *covering edge*.

The DAG implies certain conditional independencies on the variables. For example, each variable is independent of its non-descendants conditioned on its parents (Koller & Friedman, 2009). More generally, the notion of *d-separation* is defined to capture all conditional independencies imposed by the DAG (Definition 5 in the appendix). Let $\mathcal{I}(P)$ be the set of all conditional independencies satisfied by the joint distribution of variables $\mathcal{X}$. Back to the aforementioned factorization problem, it can be shown that every d-separation in the DAG that corresponds to the factorization is also satisfied by the distribution $P$. So DAG $\mathcal{G}$ corresponding to the desired factorization must fulfill the so-called *Markov condition (MC)*, that is, $\mathcal{I}(\mathcal{G}) \subseteq \mathcal{I}(P)$. A DAG $\mathcal{G}$ satisfying MC is known as an *I-map* for $P$. On the other hand, we are interested in the "simplest" factorization, that is, the "sparsest" DAG. This is captured by the notion of *minimal I-map*, which is a DAG that is an I-map for $P$ but not if any of its edges are deleted. We make the more restrictive yet common assumption that $\mathcal{I}(P) \subseteq \mathcal{I}(\mathcal{G})$. The distribution $P$ is said to be *faithful* to the DAG $\mathcal{G}$ if it satisfies this assumption. We assume that there exists a DAG $\mathcal{G}$ that satisfies both the Markovness and faithfulness assumptions for the distribution $P$. Such a DAG is called a *P-map* for $P$, also referred to as the *true DAG*.

**Assumption 1** *There exists a DAG $\mathcal{G}$ that is a P-map for the distribution $P$, i.e., $\mathcal{I}(\mathcal{G}) = \mathcal{I}(P)$.*

The factorization problem then is to find a P-map for $P$ – a task known as *structure learning*. There is often more than one P-map for a distribution $P$, e.g., both DAGs $X \to Y$ and $Y \to X$ are a P-map for the distribution $\mathcal{I}(P) = \emptyset$. The set of all P-maps for a given distribution $P$ share the same skeleton (the undirected graph obtained by removing the orientations of the edges) and immoralities (Koller & Friedman, 2009). Thus, a *partially DAG (PDAG)*, that is a graph that can have both directed and

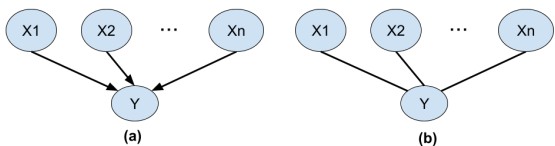

Figure 1: (a) True DAG (b) The undirected graph after testing marginal independencies by the PC algorithm.

undirected edges but does not have a directed cycle, is used to represent the set of all P-maps. More specifically, a *P-map class PDAG for $P$* is a PDAG, such that if the undirected edges are oriented in a way that do not make a directed cycle or a new immorality, the resulting DAG is a P-map for $P$.

The PC algorithm obtains a P-map class PDAG by first forming a completely-connected undirected graph over variables $\mathcal{X}$, then iteratively removing the edge between every two variables that are marginally independent, then those that are independent conditioned on a single other variable, then those that are independent conditioned on two other variables, and so on. Finally, the directions of the remaining edges are determined based on the orientation rules described in (Spirtes et al., 2000) (Algorithm 3). Consequently, the number of conditional independent (CI) tests performed by PC is at most $2\binom{N}{2}\sum_{i=0}^{N-2}\binom{N-2}{i}$ and is thus of order $\mathcal{O}(2^N)$.

The idea behind the algorithm is that two adjacent nodes in the P-map never become independent, regardless of what they are conditioned on, because they never become d-separated.

**Lemma 1** *(Based on (Pearl, 2009)) Consider random variables $\mathcal{X}$ with joint distribution $P$ that admits a P-map $\mathcal{G}$. Vertices $X$ and $Y$ are not adjacent in $\mathcal{G}$ if and only if $X \perp Y \mid \mathcal{U}$ for some $\mathcal{U} \subseteq \mathcal{X}$.*

PC is based on a sharpened version of this lemma, restricting $\mathcal{U}$ to the parents of $X$ and $Y$.

**Lemma 2** *[Lemma 3.2 in (Koller & Friedman, 2009)] Consider random variables $\mathcal{X}$ with joint distribution $P$ that admits a P-map $\mathcal{G}$. Vertices $X$ and $Y$ are not adjacent in $\mathcal{G}$ if and only if $X \perp Y \mid \mathrm{Pa}_X$ or $X \perp Y \mid \mathrm{Pa}_Y$.*

Thus, in order to remove the edge between $X$ and $Y$, only CI tests conditioned on the parents of $X$ and conditioned on the parents of $Y$ are needed. As the parents are unknown initially in the undirected graph, PC searchers over all subsets of the neighbors of $X$, denoted $\mathcal{N}_X$, and the neighbors of $Y$, denoted $\mathcal{N}_Y$, to ensure that their parents will be conditioned on during the search.

**Example 1** *Consider random variables $X_1, \ldots, X_n$ and $Y$ and assume that their joint distribution admits a P-map in the form of Fig. 1-a. Starting from the completely connected graph, PC algorithm performs marginal independence tests on pairs of variables. If first the links between the nodes $X_1, \ldots, X_n$ are checked, then the undirected structure in Fig. 1-b is obtained before checking the links between $Y$ and the other nodes. This is done by marginal independence tests between $X_i$ and $X_j$ for all $i$ and $j$, that is, $n(n-1)/2$ tests. Next, the dependence between $Y$ and the $X_i$'s is checked. Since these links are not spurious, i.e., exist in the P-map, PC performs all possible CI tests to ensure that they cannot be removed. For example, for the link $X_1 \to Y$, PC checks whether $X_1 \perp Y \mid \mathcal{U}$ holds for all combinations $\mathcal{U} \subseteq \{X_2, \ldots, X_n\}$ from zero to $n-1$ variables, resulting in $\mathcal{O}(2^n)$ tests.*

**Example 2** *Consider variables $\mathcal{X} = \{X_1, \ldots, X_{11}\}$ whose joint distribution admits the P-map $\mathcal{G}$ in Fig. 2-a. Suppose that we start the PC algorithm from the undirected graph in Fig. 2-b and in particular, we are interested in determining whether the link between $X_5$ and $X_8$ also exists in the P-map. To this end, PC checks the CI's $X_5 \perp X_8 \mid \mathcal{U}$ for all subsets $\mathcal{U} \subseteq \mathcal{N}_{X_5} = \{X_{12}, X_{13}, X_2, X_3, X_6, X_7, X_{10}, X_9, X_{15}\}$ and $\mathcal{U} \subseteq \mathcal{N}_{X_8} = \{X_4, X_3, X_6, X_7, X_{11}, X_{17}\}$.*

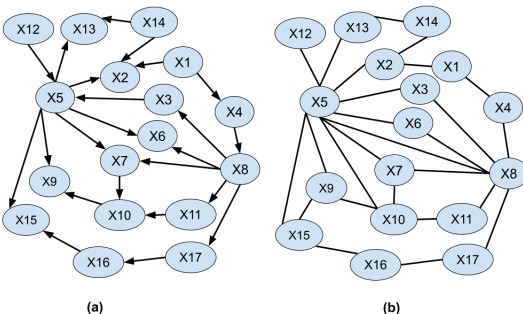

Figure 2: (a) True DAG (b) Undirected graph at some point in the PC algorithm

# 3 THE PATH-DRIVEN INDEPENDENCE TESTING (PIT) ALGORITHM

## 3.1 THE IDEA

Is it possible to reduce the search over all subsets $\mathcal{U}$ of the neighbors $\mathcal{N}_X$ and $\mathcal{N}_Y$ in the PC algorithm? Lemma 2 is conservative in the sense that among the parents of $X$ (resp. $Y$), some may not be even indirectly connected to $Y$ (resp. $X$). When checking the link $X_1 \to Y$ in Example 1, node $Y$ has many parents, but none are connected to $X_1$. Therefore, $X_1$ and $Y$ are not d-separated in the true DAG conditioned on any combination of the parents of $Y$. Similarly, $X_1$ has no parent. Hence, an efficient algorithm should perform only $n$ marginal independent tests between $Y$ and each of $X_1, \ldots, X_n$ once the undirected structure in Fig. 1-b is reached. This observation suggests that the parents $\mathrm{Pa}_X$ (resp. $\mathrm{Pa}_Y$) in Lemma 2 should be limited to those that are connected to the other node $Y$ (resp. $X$) by some path. We define the notion of *covering paths* accordingly.

**Definition 1 (Covering path)** *A* covering path/trail *for nodes $X$ and $Y$ is a trail between $X$ and $Y$ of length at least two. A covering path is* short *if the length is exactly two and* long *otherwise. The set of all covering, short-covering, and long-covering paths between $X$ and $Y$ are denoted by $\mathcal{T}_{XY}$, $\mathcal{T}_{XY}^s$, and $\mathcal{T}_{XY}^l$.*

Rather than the whole parents $\mathrm{Pa}_X$, one may condition only on those that belong to the interior of the covering paths between $X$ and $Y$, which we refer to as the *essential parents of $X$ (with respect to $Y$)*, denoted by $\mathrm{Pa}_{X,Y} = \mathrm{Pa}_X \cap \{\mathrm{int}(\mathcal{T}) : \mathcal{T} \in \mathcal{T}_{XY}\}$. This is the same as those parents of $X$ that can reach node $Y$ (excluding $Y$ itself), i.e., $\mathrm{Pa}_{X,Y} = \mathrm{Pa}_X \cap \mathcal{R}_Y \setminus \{Y\}$. In Figure 1-a, $\mathrm{Pa}_{X_i,Y} = \emptyset$ for all $i$, and in Figure 2-a, $\mathrm{Pa}_{X_5 X_8} = \{X_3\}$.

A second conservative aspect of PC is in the way Lemma 2 is used. As discussed earlier, since the parents are not a priori known in the undirected graph, PC may search through eventually all possible combinations of $\mathcal{N}_X$. However, to reduce the number of possible combinations, is it possible to always include some nodes in the search, namely, to always include a neighbor, say $Z \in \mathcal{N}_X$, in the CI tests as $X \perp Y | Z, \mathcal{U}$? Generally, no, because node $Z$ may be the center node of a collider on a trail connecting $X$ and $Y$. Then observing $Z$ may activate the trail and prevent $X$ and $Y$ from becoming independent. An example is in Fig. 2-a in Example 2, where $X_2$ is not a parent of $X_5$ and observing it makes the trail $(X_5, X_2, X_1, X_4, X_8)$ active. Nevertheless, there are situations where observing some nodes does not make the connecting trails active. For example, if instead of just $X_2$, we observe all of the interior nodes of the trail, i.e., $X_2, X_1$, and $X_4$, then the trail becomes inactive. So we can include these nodes in the conditioned nodes in all of the CI tests. That is, rather than checking $X_5 \perp X_8 | \mathcal{U}$ for all $\mathcal{U} \subseteq \mathcal{N}_X$ and $\mathcal{U} \subseteq \mathcal{N}_Y$, we can check $X_5 \perp X_8 | X_2, X_1, X_4, \mathcal{U}'$ for all $\mathcal{U}' \subseteq \mathcal{N}_X \setminus \{X_2\}$ and $\mathcal{U}' \subseteq \mathcal{N}_Y \setminus \{X_4\}$, reducing the search space. The following result illustrates the nodes that can be always observed.

**Lemma 3** *Consider a DAG $\mathcal{G}$ and let $\mathcal{T} \in \mathcal{T}_{XY}^l$ be a long-covering trail between nodes $X$ and $Y$ in $\mathcal{G}$. Then the followings hold: (i) there is an interior node of $\mathcal{T}$ that if observed, makes the trail inactive; (ii) if two adjacent interior nodes of $\mathcal{T}$ are observed, then the trail is inactive, regardless of whether some of the remaining nodes are observed; (iii) if all interior nodes of $\mathcal{T}$ are observed, the trail becomes inactive.*

Lemma 3 suggests that either all nodes or any pair of adjacent nodes of the interior of a long-covering path should be always observed in the CI tests. However, observing the interior of the long-covering path may activate some other trails, such as a collider between $X$ and $Y$, which should be avoided. More specifically, an interior node of a long-covering path may be a collider node or a descendant in some short-covering path. For example, node $X_{10}$ in Fig. 2-a belongs to the long-covering path $(X_5, X_9, X_{10}, X_{11}, X_8)$ but is a descendant of $X_7$ that forms a collider with $X_5$ and $X_8$. Observing $X_{10}$ activates the collider and renders $X_5$ and $X_8$ dependent.

**Definition 2 (Potentially trigger node)** *Consider a graph $\mathcal{G}$ with nodes $\mathcal{X}$ and let $X, Y \in \mathcal{X}$. A potentially trigger node of $X$ and $Y$ is a node that is connected to the interior node of some short-covering path of $X$ and $Y$ via a path that does not include $X$ and $Y$. A path is potentially trigger if it contains a potentially trigger node. The set of all potentially trigger nodes of $X$ and $Y$ is denoted by $\mathcal{W}_{XY}$.*

So $X_{10}$ is a potentially trigger node of the long-covering path $(X_5, X_9, X_{10}, X_{11}, X_8)$. Thus, in Fig. 2-b when performing the CI tests over the neighbors $\mathcal{N}_X$, we cannot set $X_{10}$, and consequently the interior of the covering path $(X_5, X_9, X_{10}, X_{11}, X_8)$, to be always observed. The definition is not limited to the nodes connected to colliders and covers all nodes connected to any short-covering trail. The connected nodes may or may not activate a collider, and hence, the term "potentially" trigger. The reason for this uncertainty is that in the undirected graph used in the structure learning algorithm, the edge directions and hence colliders are unknown. The remaining covering paths do not include potentially trigger nodes and are defined in the following. A *connected component* of a graph is a connected subgraph (every pair of nodes of which are connected by some trail) that itself is not a subgraph of any other connected subgraph. Given graph $\mathcal{G}$ with node set $\mathcal{X}$, the *graph $\mathcal{G}[\mathcal{V}]$ induced by the subset $\mathcal{V} \subseteq \mathcal{X}$* is a graph with node set $\mathcal{V}$ where two nodes are linked if and only if they are linked in the original graph $\mathcal{G}$.

**Definition 3 (Blindly blockable)** *Consider graph $\mathcal{G}$ with nodes $\mathcal{X}$ and let $X, Y \in \mathcal{X}$. A (blindly) blockable path between $X$ and $Y$ is a long covering path without potentially trigger nodes, that is, a path $\mathcal{T} \in \mathcal{T}^l_{XY}$ such that $\mathcal{T} \cap \mathcal{W}_{XY} = \emptyset$. The set of (blindly) blockable paths between $X$ and $Y$ is denoted by $\mathcal{T}^b_{XY}$. The maximal blockable set of $X$ and $Y$ is defined by*

$$\mathcal{C}_{XY} \triangleq \mathcal{X} \setminus (\mathcal{W}_{XY} \cup \{X, Y\}).$$

*The maximal blockable neighborhood of $X$ relative to $Y$ is defined by*

$$\bar{\mathcal{B}}_{XY} = \{Z_1, Z_2 \in \mathcal{T} : Z_1 \in \mathcal{N}_X, Z_2 \in \mathcal{N}_{Z_1}, \mathcal{T} \in \mathcal{T}^b_{XY}\}.$$

*The (minimal) blockable neighborhood of $X$ relative to $Y$, denoted $\mathcal{B}_{XY}$, is the smallest subset of $\bar{\mathcal{B}}_{XY}$ such that at least two consecutive nodes of every blockable path is included in $\mathcal{B}_{XY}$. A connected blockable neighborhood of $X$ relative to $Y$ is a connected component of the graph $\mathcal{G}[\mathcal{B}_{XY}]$. An off-path blockable set of $X$ and $Y$ is a subset of $\mathcal{C}_{XY} \setminus \mathcal{T}^b_{XY}$.*

In Figure 2-a, the blockable paths between $X_5$ and $X_8$ are $(X_5, X_{13}, X_{14}, X_2, X_1, X_4, X_8)$, $(X_5, X_2, X_1, X_4, X_8)$, and $(X_5, X_{15}, X_{16}, X_{17}, X_8)$. The maximal blockable set is the complement node set of $X$ and $Y$ and the potentially trigger nodes in between, i.e., $\mathcal{C}_{X_5 X_8} = \{X_{12}, X_{13}, X_{14}, X_2, X_1, X_4, X_{15}, X_{16}, X_{17}\}$. In view of Lemma 3, this is the maximum set of nodes that can be "blindly" conditioned on without rendering $X$ and $Y$ independent. The maximal blockable neighborhood is the union of the neighbors and the neighbors of the neighbors of $X$ on all non-potentially trigger covering paths between $X$ and $Y$, i.e., $\bar{\mathcal{B}}_{X_5 X_8} = \{X_{13}, X_{14}, X_2, X_1, X_{15}, X_{16}\}$. The idea is that according to Lemma 3, by observing two adjacent nodes on the blockable paths, they become inactive regardless of whether one of the two nodes is a collider node (and hence the term "blindly blockable"). The (minimal) blockable neighborhood is the same as the maximal blockable neighborhood, where some "redundant" paths that start from $X$ and end at a neighbor of $X$, say $Z_1$, are ignored, because they do not need to be blocked if $Z_1$ and its neighbor $Z_2$ are blocked. The path $(X_5, X_{13}, X_{14}, X_2)$ is redundant, yielding $\mathcal{B}_{X_5 X_8} = \{X_2, X_1, X_{15}, X_{16}\}$. The connected blockable neighborhoods are $\mathcal{B}^1_{X_5 X_8} = \{X_2, X_1\}$ and $\mathcal{B}^2_{X_5 X_8} = \{X_{15}, X_{16}\}$. An off-path blockable set is a collection of blindly blockable nodes that are not on any blindly blockable path between $X$ and $Y$, i.e., $\mathcal{O}_{X_5 X_8} = \{X_{12}\}$.

Now, based on our discussion on the notion of covering paths (Definition 1), $\mathrm{Pa}_X$ in Lemma 2 can be restricted to the essential parents $\mathrm{Pa}_{X,Y}$. On the other hand, some essential parents belong to

a blindly blockable path and may be excluded from the search space provided that the blockable neighborhood $\mathcal{B}_{XY}$ is observed. Indeed, one can show that $X \perp Y \mid \mathrm{Pa}_{X,Y} \cup \mathcal{B}_{XY}$, suggesting that the structure learning algorithm can search through $\mathcal{N}_X \setminus \mathcal{B}_{XY}$ to find $\mathcal{U} = \mathrm{Pa}_{X,Y} \setminus \mathcal{B}_{XY}$ and then the CI test $X \perp Y \mid \mathcal{U} \cup \mathcal{B}_{XY}$ can be performed.

However, in a structure learning algorithm that starts with a complete graph and iteratively deletes edges by checking CI tests, the set $\mathcal{B}_{XY}$ is not initially identified, because neither is $\mathcal{W}_{XY}$. Given graph $\mathcal{G}$ that is obtained at a specified iteration, we use superscript $\mathcal{G}$ for the sets that are defined based on $\mathcal{G}$, e.g., $\mathcal{W}_{XY}^{\mathcal{G}}$ and $\mathcal{B}_{XY}^{\mathcal{G}}$. In the beginning, the set of potentially trigger nodes $\mathcal{W}_{XY}^{\mathcal{G}}$ consists of all nodes except for $X$ and $Y$ and no node path is blindly blockable, resulting in $\mathcal{B}_{XY}^{\mathcal{G}} = \emptyset$. Once some edges are removed over the iterations, $\mathcal{W}_{XY}^{\mathcal{G}}$ shrinks and $\mathcal{B}_{XY}^{\mathcal{G}}$ equals the union of some connected blockable neighborhoods of $X$ relative to $Y$, denoted $\mathcal{B}_{XY}^C$, and some superfluous blockable neighborhoods that are actually an off-path blockable set, i.e., $\mathcal{O}_{XY}$, due to some superfluous edges to be removed in future iterations. We, therefore, modify the parents $\mathrm{Pa}_X$ in Lemma 2 to the following.

**Definition 4 (Separator)** *Consider DAG $\mathcal{G}$ with nodes $\mathcal{X}$ and let $X, Y \in \mathcal{X}$. A (path-driven) separator for $X$ relative to $Y$ is defined by*

$$\mathcal{M}_{XY}^* \triangleq \mathrm{Pa}_{X,Y} \cup \mathcal{B}_{XY}^C \cup \mathcal{O}_{XY}$$

*where $\mathcal{B}_{XY}^C$ is the union of an arbitrary (possibly empty) collection of connected blockable neighborhoods of $X$ relative to $Y$, and $\mathcal{O}_{XY}$ is an off-path blockable set of $X$ and $Y$.*

In Fig. 1-a, for all $i$, $\mathcal{M}_{X_iY}^*$ can be $\emptyset$ or more generally, any subset of $\{X_1, \ldots, X_n\} \setminus \{X_i\}$ which is an off-path blockable set. In Fig. 2-a, the possible values for $\mathcal{M}_{X_5 X_8}^*$ are $\{X_3\}$, $\{X_3, X_2, X_1\}$, $\{X_3, X_{12}\}$, $\{X_3, X_2, X_1, X_{12}\}$, $\{X_3, X_{15}, X_{16}\}$, and $\{X_3, X_{15}, X_{16}, X_{12}\}$.

The separator is asymmetric with respect to $X$ and $Y$, i.e., $\mathcal{M}_{XY}^* \neq \mathcal{M}_{YX}^*$. The following counterpart to Lemma 2 implies that it is necessary and sufficient for two non-adjacent variables to be independent, conditioned on their separators. The proofs are provided in the Appendix.

**Lemma 4** *Consider random variables $\mathcal{X}$ with joint distribution $P$ that admits a P-map $\mathcal{G}$. Nodes $X$ and $Y$ are not adjacent in $\mathcal{G}$ if and only if $X \perp Y \mid \mathcal{M}_{XY}^*$ for every separator $\mathcal{M}_{XY}^*$ or $X \perp Y \mid \mathcal{M}_{YX}^*$ for every separator $\mathcal{M}_{YX}^*$.*

### 3.2 THE ALGORITHMS

We develop three sound and complete structure-learning algorithms. The idea is to find the essential parents $\mathrm{Pa}_{X,Y}$ (or $\mathrm{Pa}_{Y,X}$) for every non-adjacent pair of nodes $X$ and $Y$. Edge directions are unknown in the undirected graph, preventing us to know the parents of $X$ and $Y$. Thus, similar to PC, where $\mathrm{Pa}_X$ was extended to $\mathcal{N}_X^{\mathcal{G}}$, we extend $\mathrm{Pa}_{X,Y}$ to $\mathcal{N}_{X,Y}^{\mathcal{G}}$, i.e., neighbors of $X$ that have a path to $Y$, and search through its subsets. That is to restrict the candidate conditioning set $\mathcal{U}$ in the PC algorithm to only the neighbors of $X$ that have a path to $Y$ when verifying the link between $X$ and $Y$. The result is the Path-Driven Independence Testing (PIT) Algorithm(PIT )–Algorithm 1.

Next is Algorithm 2, which is based on Lemma 4. Those parents that are included in the blockable paths can be always observed and are not required to be searched through. So instead of $\mathcal{N}_{X,Y}^{\mathcal{G}}$, we only need to search through $\mathcal{N}_{X,Y}^{\mathcal{G}} \setminus \mathcal{B}_{XY}$. As discussed earlier, $\mathcal{B}_{XY}$ is unknown from the start though. Instead, at every iteration $m$, $\mathcal{B}_{XY}^{\mathcal{G}}$ is available, that is the blockable set in the undirected graph $\mathcal{G}$ obtained before iteration $m$. So we search through $\mathcal{N}_{X,Y}^{\mathcal{G}} \setminus \mathcal{B}_{XY}^{\mathcal{G}}$ which can be shown to be the same as $\mathcal{N}_X^{\mathcal{G}} \cap \mathcal{W}_{XY}^{\mathcal{G}}$. The sets $\mathcal{W}_{XY}^{\mathcal{G}}$ and $\mathcal{B}_{XY}^{\mathcal{G}}$ are obtained from Algorithm 4. The adjacency set of $X$ in graph $\mathcal{G}$ is $\mathrm{Adj}(\mathcal{G}, X)$, and $\mathcal{R}_Y^{\mathcal{G}}$ is the reachable set of node $Y$ in graph $\mathcal{G}$.

Although the intuition of Algorithm 2 is based on covering paths, which are expensive to find– generally of order $\mathcal{O}(2^N)$–the algorithm does not explicitly find covering paths. When investigating the existence of a certain link $X - Y$, instead of finding every long-covering path between $X$ and $Y$ to see whether it is blockable by checking if any of its interior nodes is connected by some trail to the center node of a short covering path between $X$ and $Y$, the following can be done (Algorithm 4): First, find all interior nodes $Z$ of the short-covering paths. This is done by simply taking the

---

**Algorithm 1:** The Path-Driven Independence Testing (PIT) Algorithm

---

**Input:** A set of variables $\mathcal{X}$ and their joint probability distribution $P$
**Output:** A partially directed acyclic graph

---

1 Form the complete undirected graph $\mathcal{G}$ over nodes $\mathcal{X}$;
2 $\mathrm{Sepset}(X, Y) = \emptyset$ for all $X, Y \in \mathcal{X}$;
3 $m = 0$
4 **while** *maximum node degree in $\mathcal{G}$ is greater than $m$* **do**
5      **for** $X \in \mathcal{X}$
6          **for** $Y \in \mathrm{Adj}(\mathcal{G}, X)$
7              **for** $\mathcal{U} \subseteq \mathcal{R}_Y^{\mathcal{G} \setminus \{X\}} \cap \mathrm{Adj}(\mathcal{G}, X)$ *and* $|\mathcal{U}| = m$
8                  **if** $X \perp Y \mid \mathcal{U}$
9                      Remove the edge $X - Y$ from $\mathcal{G}$;
10                      $\mathrm{Sepset}(X, Y) \leftarrow \mathcal{U}$;
11      $m = m + 1$;
12 Orient the edges using the orientation rules in (Spirtes et al., 2000).

---

**Algorithm 2:** The Blocked-Path Driven Independence Testing (BPIT) Algorithm

---

**Input:** A set of variables $\mathcal{X}$ and their joint probability distribution $P$
**Output:** A partially directed acyclic graph

---

1 Form the complete undirected graph $\mathcal{G}$ over nodes $\mathcal{X}$;
2 $\mathrm{Sepset}(X, Y) = \emptyset$ for all $X, Y \in \mathcal{X}$;
3 $m = 0$
4 **while** *maximum node degree in $\mathcal{G}$ is greater than $m$* **do**
5      **for** $X \in \mathcal{X}$
6          **for** $Y \in \mathrm{Adj}(\mathcal{G}, X)$
7              **for** $\mathcal{U} \subseteq \mathcal{W}_{XY}^{\mathcal{G}} \cap \mathrm{Adj}(\mathcal{G}, X)$ *and* $|\mathcal{U}| \in [m - |\mathcal{B}_{XY}^{\mathcal{G}} \cap \mathrm{Adj}(\mathcal{G}, X)|, m]$
8                  **if** $X \perp Y \mid \mathcal{B}_{XY}^{\mathcal{G}} \cup \mathcal{U}$
9                      Remove the edge $X - Y$ from $\mathcal{G}$;
10                      $\mathrm{Sepset}(X, Y) \leftarrow (\mathcal{B}_{XY}^{\mathcal{G}} \cap \mathrm{Adj}(\mathcal{G}, X)) \cup \mathcal{U}$;
11      $m = m + 1$;
12 Orient the edges using the orientation rules in (Spirtes et al., 2000).

---

intersection $\mathcal{N}_X^{\mathcal{G}} \cap \mathcal{N}_Y^{\mathcal{G}}$. Next, iteratively find and remove the reachable set $\mathcal{R}_Z$ of each such node $Z \in \mathcal{N}_X^{\mathcal{G}} \cap \mathcal{N}_Y^{\mathcal{G}}$ from the graph $\mathcal{G} \setminus \{X, Y\}$. Finding the reachable set of a node in a graph with $N$ nodes can be done by for example using the adjacency matrix in $\mathcal{O}(N^2)$ (Gersting, 2006). The removed nodes form $\mathcal{W}_{XY}^{\mathcal{G}}$ and what remains is the maximal blockable set $\mathcal{C}_{XY}^{\mathcal{G}}$. For the blockable neighborhoods, we first find the reachable nodes of $Y$ when the potentially trigger and the neighbors of $X$ are removed from the graph, resulting in $\mathcal{R}_Y'$. Then for each neighbor of $X$ that is in the blockable set $\mathcal{C}_{XY}^{\mathcal{G}}$, if it shares a neighbor in $\mathcal{R}_Y'$, then it has a path to $Y$ that does not pass another neighbor of $X$. Hence, they can both be blocked.

A more efficient version of Algorithm 2 is Algorithm 5, where fewer number of candidate $\mathcal{U}$ sets are searched. The idea is that if the set $\mathcal{W}_{XY}^{\mathcal{G}}$ has not changed at a certain iteration $m$ compared to the last iteration $m - 1$, then the set $\mathcal{B}_{XY}^{\mathcal{G}}$ has not changed (except that it could have become smaller to exclude some of the nodes in $\mathcal{O}_{XY}^{\mathcal{G}}$). Then testing $\mathcal{U}$'s with a cardinality smaller than $m$ would be redundant as they were already checked in the previous iterations. In general, at iteration $m$, we need to search for candidate $\mathcal{U}$'s of size less than $m$ only if new variables are included in the set $\mathcal{B}_{XY}^{\mathcal{G}}$ compared to the last iteration $m - 1$. These new variables come from $\mathcal{W}_{XY}^{\mathcal{G}}$; hence, when searching through the candidates $\mathcal{U}$, we should consider the case where these new variables are already included in $\mathcal{U}$, and hence, start the search for $\mathcal{U}$ sets with size $m$ minus the number of the new variables, resulting in Algorithm 5.

### 3.3 EXAMPLES

The following example illustrates how the new algorithm reduces the number of required CI tests to obtain the true DAG compared to PC. The second example illustrates the usage of the blockable paths.

**Example 1 (revisited)** *By using Algorithm 2 to obtain the true DAG in Fig. 1-a, first, a complete undirected graph is constructed, and then similar to PC, the graph in Fig. 2-b is obtained after performing the iterations for $m = 0$. However, then for checking each of the links $Y - X_i$, $i \in \{1, \ldots, n\}$, No covering path exists for each pair of variables $Y$ and $X_i$, i.e.,*

$$\mathcal{T}^s_{X_i Y} = \mathcal{T}^l_{X_i Y} = \mathcal{T}^b_{X_i Y} = \mathcal{B}_{X_i Y} = \mathcal{W}_{X_i Y} = \emptyset.$$

*Consequently, the edge deletion part terminates. The resulting number of CI tests is $\mathcal{O}(n^3)$ while for the PC algorithm, it was $\mathcal{O}(2^{n-1})$.*

**Example 2 (revisited)** *Regarding $X_5 - X_8$ in Fig. 2-b, we have $\mathcal{W}_{X_5 X_8} = \{X_3, X_6, X_7, X_9, X_{10}, X_{11}, X_{15}, X_{16}, X_{17}\}$, $\mathcal{B}_{X_5 X_8} = \{X_2, X_1\}$. Thus, $\mathcal{W}_{X_5 X_8} \cap \text{Adj}(\mathcal{G}, X) = \{X_3, X_6, X_7, X_9, X_{15}\}$, $\mathcal{B}_{X_5 X_8} \cap \text{Adj}(\mathcal{G}, X) = \{X_2\}$. So by checking all subsets $\mathcal{U} \in \{X_3, X_6, X_7, X_9, X_{15}\}$ where the size of $\mathcal{U}$ belongs to the interval $[m - 1, m]$, a separator is guaranteed to be found for $m = 1$ or $2$, e.g., $\mathcal{M}^*_{X_5 X_8} = \{X_3\}$ or $\{X_3, X_2\}$.*

### 3.4 SOUNDNESS, COMPLETENESS, AND COMPLEXITY

**Theorem 1** *Consider random variables $\mathcal{X}$ with joint distribution $P$ that admits a P-map. The output graph $\hat{\mathcal{G}}$ of each of the Algorithms 1, 2, and 5 is a P-map class PDAG for $P$.*

The complexity of Algorithms 1, 2, and 3 is of order $2^N$, as in a fully connected graph, all edges must be tested using all possible CI tests. However, the number of CI tests in these Algorithm is always less than or equal to that of the standard PC algorithm (Propositions 1 and 2). Table 1 compares these algorithms across various graph structures, including naive Bayes (where one node is connected to all others), star (the same structure but with reversed edges), and a forest skeleton (a DAG with the skeleton without cycles). Here, $d$ represents the maximum number of neighbors, while $s$ denotes the maximum number of neighbors that have a path to a common node, where $s \leq d$.

**Proposition 1** *Consider random variables $\mathcal{X} = \{X_1, \ldots, X_N\}$ with joint probability distribution $P$ that admits a P-map. Assume that for each $m$, subsets $\mathcal{U}$ in both Algorithms 1, 2, and 3 are tested according to the lexicographical order. Then the total number of CI tests performed by Algorithm 2 is less than or equal to that in Algorithm 3.*

Table 1: Number of CI tests for special structures.

| METHOD | BOUNDED DEGREE | STAR | NAIVE BAYES | FOREST SKELETON |
|--------|----------------|------|-------------|-----------------|
| PC | $N^{d+2}$ | $2^{N-1}$ | $2^{N-1}$ | $N^{d+2}$ |
| PIT | $N^{s+2}$ | $N^2$ | $N^3$ | $N^3$ |

## 4 EXPERIMENTS

We compared the performance of Algorithm 1 (PIT) and its variations, Algorithm 2 (BPIT), and Algorithm 5 (Opt. BPIT), with PC, PC-stable (Colombo & Maathuis, 2014), Hill-climbing (HC), and Tabu algorithms on the datasets ("true" DAGs) ASIA (Lauritzen & Spiegelhalter, 1988), CANCER (Korb & Nicholson, 2010a), EARTHQUAKE (Korb & Nicholson, 2010b), ALARM (Beinlich et al., 1989), INSURANCE (Binder et al., 1997), CHILD (Spiegelhalter & Cowell, 1992), WATER (Jensen et al., 1989), HAILFINDER (Abramson et al., 1996), MUNIN (Andreassen et al., 1989), ANDES (Conati et al., 1997), and DIABETES (Andreassen et al., 1991). The computations were performed on a system with 2 xAMD Rome 7532@ 2.4GHz 256M cache. For each true DAG, a dataset was generated by sampling 10,000 (resp. 1,000 and 10) times from the DAG and its conditional probability

distributions, resulting in 16 datasets, each containing 10,000 data instances (resp. 1000 and 10). This dataset was passed to the learning algorithms to estimate the true DAG. For each dataset and each algorithm, we reported runtime (Table 2), the number of CI tests (Table 3), and the structural Hamming distance, that is, the number of incorrect edges, either missing or extra, compared to the true graph and divided by the total number of edges in the true DAG (Table 4). The results for 100 and 1000 samples are in the Appendix.

PIT was a clear winner. According to the Wilcoxon signed ranked test, PIT was significantly faster (up to 7 times faster; $p = .029$) and conducted significantly fewer CI tests ($p < .001$) compared to PC and, in turn, PC-stable. PIT was also often more accurate or no worse compared to PC and PC-stable, although there was no significance difference ($p = .61$, $p = .78$). The results for 100 and 1000 samples complemented the above results: PIT was significantly better in terms of CI tests and accuracy, but no significant difference in terms of runtime. PIT was also significantly faster than BPIT and Opt. BPIT and comparable in terms of the number of CI tests and accuracy.

BPIT and Opt. BPIT performed significantly fewer CI tests compared to PC and PC-stable, and while they were comparable in terms of speed, they exhibited significantly lower accuracy. Additionally, we evaluated these methods with an oracle for the CI tests (Table 14 and 15). Both methods were significantly faster than PC and PC-stable for datasets with 100 and 1000 samples, with no significant speed difference observed for datasets with 10,000 samples. This underscores the bottleneck caused by the large conditional sets used in the CI tests for these two methods.

We additionally compared the algorithms with two score based methods, Hill Climbing and Tabu, with maximum number of iterations set to 1,000,000 and $\epsilon = .0001$. We increased the number of initial conditions from 1 to 10 to 100 to ensure that the algorithms spent at least as much time on local searches as that of PIT for each dataset, i.e., the same runtime. However, still these methods were always much less accurate than PIT (Table 12).

Table 2: Run time (second)

| Dataset | BPIT | Opt. BPIT | PIT | PC | PC-stable |
|---------|------|-----------|-----|-----|-----------|
| Earthquake | 0.064 | 0.07 | **0.063** | 0.198 | 0.226 |
| Cancer | 0.033 | 0.038 | **0.032** | 0.191 | 0.203 |
| Survey | 0.138 | 0.147 | **0.077** | 0.28 | 0.291 |
| Asia | 0.336 | 0.36 | **0.335** | 0.472 | 0.65 |
| Sachs | 26.1 | 26.5 | **25.7** | 25.8 | 30.8 |
| Child | 12.4 | 12.9 | **10.7** | 47.5 | 67 |
| Insurance | 69.2 | 96.9 | **55.9** | 83.5 | 106.4 |
| Water | 16.2 | 16.3 | **6.8** | 6.98 | 11.1 |
| Mildew | 481.1 | 477 | **153.6** | 184.6 | 284.4 |
| Alarm | 30 | 30.7 | **25.1** | 27.8 | 53 |
| Hailfinder | 32671 | 33020 | **27845** | 30604 | 53834 |
| Hepar2 | 315 | 331 | **84.2** | 741 | 1033 |
| Win95pts | 132 | 151 | **87.2** | 121 | 179 |
| Munin | 17602 | 20421 | 14758 | **14042** | 28262 |
| Andes | 6199 | 8859 | 1996 | **1446** | 1831 |
| Diabetes | 245345 | 291704 | **163769** | $> 1209600$ | $> 1209600$ |

## 5 Conclusion

We introduced PIT as efficient enhancements to the PC algorithm for Bayesian network structure learning. By restricting the conditioning sets to neighbors that have paths to the target variable when verifying a link, PIT substantially reduces the number of required conditional independence (CI) tests. This straightforward modification leads to significant improvements in both speed and accuracy.

The algorithms are both sound and complete, ensuring the identification of the correct P-map under ideal conditions–something many fast algorithms compromise on. Unlike score-based methods that rely on local search approximations, PIT delivers high accuracy without relying on extensive CI testing. Moreover, PIT achieves these performance gains without necessitating additional computational resources, such as distributed or parallel computing. Instead, it complements these methods by seamlessly integrating into existing parallel or distributed frameworks where local learning algorithms are employed.

Table 3: Number of CI tests.

| DATASET | BPIT | OPT. BPIT | PIT | PC | PC-STABLE |
|---------|------|-----------|-----|-----|-----------|
| EARTHQUAKE | **31** | **31** | **31** | 57 | 64 |
| CANCER | **17** | **17** | **17** | 45 | 46 |
| SURVEY | **29** | **29** | **29** | 55 | 55 |
| ASIA | **96** | **96** | 102 | 124 | 128 |
| SACHS | **971** | **971** | 972 | **971** | 1248 |
| CHILD | **1102** | **1120** | 1262 | 2124 | 3213 |
| INSURANCE | 4342 | **4339** | 4604 | 5078 | 6588 |
| WATER | 1274 | **1272** | 1293 | 1346 | 1747 |
| MILDEW | **3345** | 3359 | 3464 | 3629 | 4969 |
| ALARM | 2845 | **2837** | 2933 | 3283 | 5293 |
| HAILFINDER | 117437 | 116979 | **107143** | 117198 | 174944 |
| HEPAR2 | 5487 | **5485** | 7020 | 23202 | 29320 |
| WIN95PTS | 9660 | **9646** | 10184 | 12501 | 17700 |
| MUNIN | 443405 | 446327 | **439255** | 448460 | 927403 |
| ANDES | **41557** | 66216 | 42172 | 68375 | 80914 |
| DIABETES | 1816592 | 1813867 | **1812709** | – | – |

Table 4: Structural Hamming Distance divided by the total number of edges (Percent)

| DATASET | BPIT | OPT. BPIT | PIT | PC | PC-STABLE |
|---------|------|-----------|-----|-----|-----------|
| EARTHQUAKE | 0 | 0 | 0 | 0 | 0 |
| CANCER | 0 | 0 | 0 | 0 | 0 |
| SURVEY | 0 | 0 | 0 | 0 | 0 |
| ASIA | 12.5 | 12.5 | 12.5 | 12.5 | 12.5 |
| SACHS | 0 | 0 | 0 | 0 | 0 |
| CHILD | 4 | 4 | **0** | 4 | 4 |
| INSURANCE | 36.5 | 32.7 | 32.7 | 32.7 | **30.8** |
| WATER | 59.1 | 59.1 | 59.1 | 59.1 | 59.1 |
| MILDEW | 45.7 | 63 | **15.2** | 17.4 | 19.6 |
| ALARM | 17.4 | 17.4 | 10.9 | **8.7** | **8.7** |
| HAILFINDER | 87.9 | 87.9 | 86.3 | 78.8 | **74.2** |
| HEPAR2 | 51.2 | 51.2 | **42.3** | 51.2 | 50.4 |
| WIN95PTS | 41.1 | 41.1 | **34.8** | 37.5 | 38.4 |
| MUNIN | 94.5 | 96.7 | 72.2 | 70 | **64.1** |
| ANDES | 41.1 | 63 | **19.2** | 19.5 | 19.8 |
| DIABETES | 107 | 108 | **56.5** | – | – |

Empirical evaluations demonstrated that PIT outperforms PC and PC-stable in terms of accuracy for small to medium-sized datasets and offers considerable speed advantages for large datasets. This dual benefit addresses the core challenges in structure learning: maintaining high accuracy with limited data and ensuring computational efficiency with large-scale data.

Advanced versions, BPIT and Opt-BPIT , build upon the path-driven approach by distinguishing between different types of connecting paths and determining if they can be "blindly" blocked. While this blind-blocking technique reduces CI tests even further, it also presents new challenges in terms of conditioning set size. However, if future CI tests can maintain accuracy while scaling linearly with conditioning size, this approach could become a game-changing advancement in the field.

Future work includes optimizing path computations to enhance algorithm efficiency and exploring robust CI testing methods that maintain accuracy with larger conditioning sets. Furthermore, integrating our approach with distributed and parallel learning techniques could enable scalability to even larger datasets, broadening the applicability of our methods in diverse real-world scenarios.

Ultimately, the contributions of PIT and BPIT strike a critical balance between precision, speed, and scalability, offering an algorithm that not only pushes the boundaries of current graph-based learning methods but also sets the stage for future innovation in causal discovery.

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

## A  APPENDIX

**Definition 5 (d-separation)** *(Koller & Friedman, 2009) Consider DAG $\mathcal{G}$ with node set $\mathcal{X}$. A trail $\mathcal{T}$ between two nodes $X, Y \in \mathcal{X}$ is* active *relative to (or given) a set of nodes $\mathcal{Z} \subseteq \mathcal{X}$ if* (i) *for each collider on $\mathcal{T}$, at least one of the descendants of the collider node is in $\mathcal{Z}$, and* (ii) *no other node on $\mathcal{T}$ is in $\mathcal{Z}$. The node subsets $\mathcal{X}_1, \mathcal{X}_2 \subseteq \mathcal{X}$ are* d-separated *given $\mathcal{Z}$, denoted $d - sep_\mathcal{G}(\mathcal{X}_1, \mathcal{X}_2 \mid \mathcal{Z})$, if there is no active trail between any node $X_1 \in \mathcal{X}_1$ and any node $X_2 \in \mathcal{X}_2$ given $\mathcal{Z}$. The set of all d-separations in $\mathcal{G}$ is denoted by $\mathcal{I}(\mathcal{G})$.*

*Proof of Lemma 3.* Since $|\mathcal{T}| \geq 2$, there is at least one non-collider triple in the trail. So the middle variable in this triple blocks the trail, which proves the first item. For the second, note that both of the adjacent nodes say $U$ and $V$, cannot be collider nodes. So $U$ and its two neighbors on $\mathcal{T}$ or $V$ and its two neighbors on $\mathcal{T}$ form a non-collider triple, which again becomes inactive once $U$ and $V$ are observed. Consequently, the trail is blocked. The third item immediately follows either of the first two. $\square$

**Proof of Lemma 4** (sufficiency) We prove the result for $\mathcal{M}_{YX}^* = \mathrm{Pa}_{X,Y} \cup \mathcal{B}_{XY}^C \cup \mathcal{O}_{XY}$, where $\mathcal{B}_{XY}^C$ and $\mathcal{O}_{XY}$ are the arbitrary sets in Definition 4. In view of Lemma 2, $X$ and $Y$ are independent conditioned on $\mathrm{Pa}_X$ or $\mathrm{Pa}_Y$. Without loss of generality assume that $X \perp Y \mid \mathrm{Pa}_X$. It follows that $X \perp Y \mid \mathrm{Pa}_{X,Y}$, because the parents of $X$ that are not reachable by $Y$ in $\mathcal{G} \setminus \{X\}$ do no activate a trial between $X$ and $Y$. Now consider a covering path $\mathcal{T}$ between $X$ and $Y$. If $\mathcal{T}$ is not connected to a node in $\mathcal{B}_{XY}^C \cup \mathcal{O}_{XY}$ by some trail in $\mathcal{G} \setminus \{X, Y\}$, then neither $\mathcal{B}_{XY}$ nor $\mathcal{O}_{XY}$ may trigger a collider on path $\mathcal{T}$. Hence, $\mathcal{T}$ is inactive once conditioned on $\mathrm{Pa}_{X,Y} \cup \mathcal{B}_{XY}^C \cup \mathcal{O}_{XY}$ as it was already inactive when conditioned on $\mathrm{Pa}_{X,Y}$. So consider the case where $\mathcal{T}$ is connected to a node in $\mathcal{B}_{XY}^C \cup \mathcal{O}_{XY}$ by some trail in $\mathcal{G} \setminus \{X, Y\}$. Then $\mathcal{T}$ is connected to $\mathcal{B}_{XY}^C$ as $\mathcal{O}_{XY}$ is off-path blockable. It follows that $\mathcal{T}$ is connected to some connected blockable neighborhood $\mathcal{B}_{XY}^c$ that is the union of $\mathcal{B}_{XY}$ and a connected component of $\mathcal{C}_{XY}$, denoted by $\mathcal{C}_{XY}^c$. Any node that is reachable by $\mathcal{B}_{XY}^c$ in $\mathcal{G} \setminus \{X, Y\}$ also belongs to the connected component $\mathcal{C}_{XY}^c$. Thus, the interior of $\mathcal{T}$ belongs to the connected component $\mathcal{C}_{XY}^c$, i.e., $\mathrm{int}(\mathcal{T}) \subseteq \mathcal{C}_{XY}^c$. So $\mathcal{T}$ is blockable, and is blocked by observing $\mathcal{B}_{XY} \cap \mathcal{T}$ in view of Lemma 3. On the other hand, $\mathcal{B}_{XY} \cap \mathcal{T} = \mathcal{B}_{XY}^c \cap \mathcal{T}$. Moreover, Lemma 3 also implies that observing additional nodes does not make the path active. So $\mathcal{T}$ is inactive if $\mathrm{Pa}_{X,Y} \cup \mathcal{B}_{XY}^C \cup \mathcal{O}_{XY}$ is observed. Therefore, every covering path between $X$ and $Y$ is inactive, implying that $X$ and $Y$ are d-separated, completing the proof as $\mathcal{G}$ is a P-map. (necessity) Should $X$ and $Y$ be independent conditioned on any set including $\mathcal{M}_{XY}^*$, they are d-separated in the P-map $\mathcal{G}$ according to Lemma 1. Thus, they cannot be linked. $\square$

**Proof of Theorem 1.** We only prove the theorem for Algorithm 2; the others can be proven similarly. Consider nodes $X, Y \in \mathcal{X}$. Should there be an edge between them in the P-map, they would not become independent conditioned on any subset of the remaining nodes according to Lemma 1. So none of the CI tests in either of the algorithms will be positive, preserving the edge.

Now consider the case where $X$ and $Y$ are not adjacent in the P-map. In view of Lemma 4, it suffices to show that at least one separator $\mathcal{M}_{XY}^*$ and at least one separator $\mathcal{M}_{YX}^*$ is found by the algorithm. We only prove the first one. Consider an arbitrary iteration $m$ of the 2 and let $\mathcal{G}$ be the graph before performing the search over the possible sets $\mathcal{U}$. Let $\mathcal{W}_{XY}, \mathcal{B}_{XY}$, and $\mathcal{C}_{XY}$ be the set of potentially trigger nodes, blockable neighbors, and maximal blockable set of the P-map, and $\mathcal{W}_{XY}^\mathcal{G}, \mathcal{B}_{XY}^\mathcal{G}$, and $\mathcal{C}_{XY}^\mathcal{G}$, the corresponding sets in $\mathcal{G}$. Since $\mathcal{G}$ is a subgraph of the P-map, $\mathcal{W}_{XY} \subseteq \mathcal{W}_{XY}^\mathcal{G}$ and in turn $\mathcal{C}_{XY}^\mathcal{G} \subseteq \mathcal{C}_{XY}$. Hence, every blockable covering path $\mathcal{T}^\mathcal{G}$ of $X$ and $Y$ in $\mathcal{G}$ satisfies $\mathrm{int}(\mathcal{T}^\mathcal{G}) \subseteq \mathcal{C}_{XY}$. Thus, if $\mathcal{T}^\mathcal{G}$ also exists in the P-map, then it is a blockable path, otherwise, all interior nodes of $\mathcal{T}^\mathcal{G}$ are off-path blockable. In the first case, $\mathrm{int}(\mathcal{T}^\mathcal{G})$ belongs to a connected component of $\mathcal{C}_{XY}$, implying $\mathcal{T}^\mathcal{G} \cap \mathcal{B}_{XY}^\mathcal{G} = \mathcal{T}^\mathcal{G} \cap \mathcal{B}_{XY}^c$ for some connected blockable neighborhood of $X$ relative to $Y$, $\mathcal{B}_{XY}^c$. So $\mathcal{B}_{XY}^\mathcal{G} = \mathcal{B}_{XY}^C \cup \mathcal{O}_{XY}$ for a union $\mathcal{B}_{XY}^C$ of some connected blockable neighborhoods of $X$ relative to $Y$ and for some off-path blockable set $\mathcal{O}_{XY} \subseteq \mathcal{C}_{XY}$. Therefore, $\mathrm{Pa}_{X,Y} \cup \mathcal{B}_{XY}^\mathcal{G}$ is a separator.

So it suffices for the set $\mathcal{U}$ to equal $\mathrm{Pa}_{X,Y} \setminus \mathcal{B}_{XY}^\mathcal{G}$ at some point in the algorithm. On the other hand, $\mathrm{Pa}_{X,Y} \setminus \mathcal{B}_{XY}^\mathcal{G}$ is the intersection of $\mathrm{Pa}_X \setminus \mathcal{B}_{XY}^\mathcal{G}$ with the reachable set of $Y$ in $\mathcal{G} \setminus \{X\}$. Therefore, $\mathrm{Pa}_{X,Y} \setminus \mathcal{B}_{XY}^\mathcal{G} = \mathrm{Pa}_X \setminus \mathcal{C}_{XY}^\mathcal{G}$, implying that $\mathrm{Pa}_{X,Y} \setminus \mathcal{B}_{XY}^\mathcal{G} = \mathrm{Pa}_X \cap \mathcal{W}_{XY}^\mathcal{G}$.

Let $|\mathrm{Pa}_{X,Y}| = k$. Consider the iteration with $m = k$ in the algorithm, and let $\mathcal{G}_k$ be the corresponding graph. It holds that $\mathrm{Pa}_{X,Y} \setminus \mathcal{B}_{XY}^{\mathcal{G}} = \mathrm{Pa}_X \cap \mathcal{W}_{XY}^{\mathcal{G}_k} \subseteq \mathcal{N}_X^{\mathcal{G}_k} \cap \mathcal{W}_{XY}^{\mathcal{G}_k}$. On the other hand, $|\mathrm{Pa}_{X,Y} \setminus \mathcal{B}_{XY}^{\mathcal{G}}| = k - |\mathrm{Pa}_{X,Y} \cap \mathcal{B}_{XY}^{\mathcal{G}_k}|$. Moreover, $|\mathrm{Pa}_{X,Y} \cap \mathcal{B}_{XY}^{\mathcal{G}_k}| \leq |\mathcal{N}_X^{\mathcal{G}_k} \cup \mathcal{B}_{XY}^{\mathcal{G}_k}|$. Hence, $k - |\mathcal{N}_X^{\mathcal{G}_k} \cap \mathcal{B}_{XY}^{\mathcal{G}_k}| \leq |\mathrm{Pa}_{X,Y} \setminus \mathcal{B}_{XY}^{\mathcal{G}_k}| \leq k$. Therefore, by searching through all subsets $\mathcal{U}$ of $\mathcal{N}_X^{\mathcal{G}_k} \cup \mathcal{B}_{XY}^{\mathcal{G}_k}$, where $k - |\mathcal{N}_X^{\mathcal{G}_k} \cap \mathcal{B}_{XY}^{\mathcal{G}_k}| \leq |\mathcal{U}| \leq k$, the desired $\mathcal{U}$ is guaranteed to be found, such that $X \perp Y \mid \mathcal{B}_{XY}^{\mathcal{G}_k} \cup \mathcal{U}$. By going through all values of $m$ the value $k$ is also found. So the edge between $X$ and $Y$ will be deleted. Therefore, the skeletons of $\hat{\mathcal{G}}$ and the P-map matches.

The orientation step of the edges in the stated algorithms is the same as the one in PC, with the difference that now $\mathrm{Sepset}(X, Y)$ may include additional nodes, i.e., those in $\mathcal{B}_{XY}$, that are not a common neighbor of $X$ and $Y$. However, this will not affect the orientations as the purpose of having the set $\mathrm{Sepset}(X, Y)$ is to check if a common neighbor belongs to it, whereas those in $\mathcal{B}_{XY}$ are not a common neighbor of $X$ and $Y$. $\qquad\square$

**Proof of Proposition 1.** We prove by contradiction. Assume on the contrary that the proposition is violated for the first time for nodes $X$ and $Y$ and iteration $m$. Consider the case where the obtained graph in both algorithms is the same so far. Also, consider the case where there is an edge between $X$ and $Y$ in the true graph. In PC, all possible $\mathcal{U}$'s of size $m$ from the set $\mathrm{Adj}(\mathcal{G}, X)$ are checked, which equals $\binom{|\mathrm{Adj}(\mathcal{G},X)|}{m}$ many candidates. In Algorithm 2, $\mathrm{Adj}(\mathcal{G}, X)$ is partitioned into $\mathcal{W}_{XY}^{\mathcal{G}}$ and $\mathrm{Adj}(\mathcal{G}, X) \setminus \mathcal{W}_{XY}^{\mathcal{G}}$. Without any optimization, the number of candidates would equal that in PC, i.e., to choose $k$ variables from $\mathcal{W}_{XY}^{\mathcal{G}}$ and $m - k$ from $\mathrm{Adj}(\mathcal{G}, X) \setminus \mathcal{W}_{XY}^{\mathcal{G}}$, i.e.,

$$\sum_k \binom{|\mathcal{W}_{XY}^{\mathcal{G}}|}{k}\binom{|\mathrm{Adj}(\mathcal{G}, X) \setminus \mathcal{W}_{XY}^{\mathcal{G}}|}{m - k} = \binom{|\mathrm{Adj}(\mathcal{G}, X)|}{m}.$$

However, the choice of $m - k$ variables from the set $\mathrm{Adj}(\mathcal{G}, X) \setminus \mathcal{W}_{XY}^{\mathcal{G}}$ does not happen in Algorithm 2; instead, the whole subset $\mathcal{B}_{XY}^{\mathcal{G}} \subseteq \mathrm{Adj}(\mathcal{G}, X) \setminus \mathcal{W}_{XY}^{\mathcal{G}}$ is observed. This reduces the number of candidate $\mathcal{U}$'s to $\sum_k \binom{|\mathcal{W}_{XY}^{\mathcal{G}}|}{k}$ which is less than or equal to $\binom{|\mathrm{Adj}(\mathcal{G},X)|}{m}$. Thus, for every iteration $m$, the number of candidate $\mathcal{U}$'s that are searched in Algorithm 2 is no more than that in PC. Now, if the graph obtained by Algorithm 2 had fewer links compared to that obtained by Algorithm 2, then $\mathcal{W}_{XY}^{\mathcal{G}}$ might have been smaller, but that does not make the total number of tests to exceed $\binom{|\mathrm{Adj}(\mathcal{G},X)|}{m}$.

Now consider the case where there is no edge between $X$ and $Y$ in the true graph. According to the proof of Theorem 1, if a certain $\mathcal{U}$ at iteration $m$ renders $X$ and $Y$ independent in PC, so does the $\mathcal{B}_{XY}^{\mathcal{G}} \cup \mathcal{U}$ in Algorithm 2. So if the edge $X - Y$ is removed by PC at iteration $m$, it will also be removed by Algorithm 2 by at most iteration $m$. On the other hand, the same lexicographical ordering is supposed for both algorithms. This means that Algorithm 2 searches through the candidates for the set $\mathcal{U}$ in the same order that PC does, just that some candidates may not be searched in Algorithm 2, and that some candidates appear in a smaller cardinality as some members belong to the set $\mathcal{B}_{XY}^{\mathcal{G}}$ which are not searched through. This means that the Algorithm 2 takes at most the same number of iterations as that in PC to find the proper $\mathcal{U}$, a contradiction. $\qquad\square$

**Proposition 2** *Consider random variables $\mathcal{X}$ with joint distribution $P$ that admits P-map $\mathcal{G}$. If $\mathcal{G}$ contains no cycle, then the total number of CI tests performed by Algorithms 1, 2, and 5 is $\mathcal{O}(N^3)$.*

**Proof of Proposition 2.** First, we prove that if between two nonadjacent nodes $X$ and $Y$ at most, there is one path in the skeleton graph, these nodes are separated by observing at most one node. If a collider node exists in the path between $X$ and $Y$, then $X$ and $Y$ are d-separated given the empty set, and if there is no collider node in the path, by observing each node $Z$ on the path $X$ and $Y$ are d-separated given $Z$. So, by checking the marginal independence test between $X$ and $Y$ or the CI test between $X$ and $Y$ given $Z$ the spurious edge between them is removed. Since there is no cycle in the skeleton, there is at most one path between any two nodes in the true DAG. Therefore, by observing at most one variable on the path between any two nodes, all spurious edges can be removed. Verifying the correctness of the remaining edges does not require additional CI tests, as there are no alternative paths with mediator nodes between any two adjacent nodes. Thus, the number of CI tests is $\mathcal{O}(N^3)$ in the worst case. $\qquad\square$

If $d$ represents the maximum number of neighbors in the true DAG, and $s$ denotes the maximum number of neighbors reachable from another node in the true DAG, the number of CI tests is bounded

by $\mathcal{O}(N^s)$ in BPIT , while the PC algorithm requires $\mathcal{O}(N^d)$ CI tests. Since $s$ is always less than or equal to $d$, it follows that $\mathcal{O}(N^s) \leq \mathcal{O}(N^d)$. Table 1 indicates the number of CI tests for some special structures.

---

**Algorithm 3:** The PC Algorithm

---

**Input:** A set of variables $\mathcal{X}$ and their joint probability distribution $P$
**Output:** A partially directed acyclic graph

1 Form the complete undirected graph $\mathcal{G}$ over nodes $\mathcal{X}$;
2 $\text{Sepset}(X, Y) = \emptyset$ for all $X, Y \in \mathcal{X}$;
3 $m = 0$
4 **while** *maximum node degree in $\mathcal{G}$ is greater than $m$* **do**
5     **for** $X \in \mathcal{X}$
6         **for** $Y \in \text{Adj}(\mathcal{G}, X)$
7             **for** $\mathcal{U} \subseteq \text{Adj}(\mathcal{G}, X) \setminus \{Y\}$ *and* $|\mathcal{U}| = m$
8                 **if** $X \perp Y \mid \mathcal{U}$
9                     Remove the edge $X - Y$ from $\mathcal{G}$;
10                     $\text{Sepset}(X, Y) \leftarrow \mathcal{U}$;
11     $m = m + 1$;
12 Orient the edges using the orientation rules in (Spirtes et al., 2000).

---

---

**Algorithm 4:** The $\mathcal{W}$, $\mathcal{B}$ and $\mathcal{C}$-finding Algorithm

---

**Input:** Undirected graph $\mathcal{G}$; nodes $X$ and $Y$
**Output:** $\mathcal{W}^{\mathcal{G}}_{XY}, \mathcal{B}^{\mathcal{G}}_{XY}, \mathcal{C}^{\mathcal{G}}_{XY}$

1 $\mathcal{W}^{\mathcal{G}}_{XY} = \mathcal{B}^{\mathcal{G}}_{XY} = \emptyset$;
2 **for** $Z \in \text{Adj}(\mathcal{G}, X) \cap \text{Adj}(\mathcal{G}, Y)$
3     $\mathcal{W}^{\mathcal{G}}_{XY} \leftarrow \mathcal{W}^{\mathcal{G}}_{XY} \cup \mathcal{R}^{\mathcal{G} \setminus \{X, Y\}}_{Z}$;        `// Reachable nodes from common neighbors of X and Y`
4 $\mathcal{C}^{\mathcal{G}}_{XY} \leftarrow \mathcal{X} \setminus (\{X, Y\} \cup \mathcal{W}^{\mathcal{G}}_{XY})$;                  `// The maximal blockable set`
5 $\mathcal{R}'_Y \leftarrow \mathcal{R}^{\mathcal{G} \setminus (\mathcal{W}^{\mathcal{G}}_{XY} \cup (\text{Adj}(\mathcal{G}, X) \setminus \{Y\}))}_{Y}$;     `// Reachable nodes from Y in the absence of trigger`
      `nodes and X's neighbors`
6 **for** $Z \in \text{Adj}(\mathcal{G}, X) \cap \mathcal{C}^{\mathcal{G}}_{XY}$            `// blockable neighbors of X`
7     **if** $\text{Adj}(\mathcal{G}[\mathcal{C}^{\mathcal{G}}_{XY}], Z) \cap \mathcal{R}'_Y \neq \emptyset$      `// that can reach Y via a blockable path`
8         $\mathcal{B}^{\mathcal{G}}_{XY} \leftarrow \mathcal{B}^{\mathcal{G}}_{XY} \cup \{Z, \text{Adj}(\mathcal{G}[\mathcal{C}^{\mathcal{G}}_{XY}], Z)\}$;      `// The blockable neighborhood`

---

---

**Algorithm 5:** The optimized Path-Driven Independence Testing Algorithm(Optimized PDIT)

---

**Input:** A set of variables $\mathcal{X}$ and their joint probability distribution $P$
**Output:** A partially directed acyclic graph

1 Form the complete undirected graph $\mathcal{G}$ over nodes $\mathcal{X}$;
2 $\text{Sepset}(X, Y) = \emptyset$ for all $X, Y \in \mathcal{X}$; $\bar{\mathcal{B}}_{XY} = \emptyset$; $\bar{\mathcal{W}}_{XY} = \emptyset$;
3 $m = 0$
4 **while** *maximum node degree in $\mathcal{G}$ is greater than $m$* **do**
5     **for** $X \in \mathcal{X}$
6         **for** $Y \in \text{Adj}(\mathcal{G}, X)$
7             **for** $\{\mathcal{U} \subseteq \mathcal{W}^{\mathcal{G}}_{XY} \cap \text{Adj}(\mathcal{G}, X)$ *and* $|\mathcal{U}| \in [m - |\mathcal{B}^{\mathcal{G}}_{XY} \cap \bar{\mathcal{W}}_{XY} \cap \text{Adj}(\mathcal{G}, X)|, m]\}$
8                 **if** $X \perp Y \mid \mathcal{B}^{\mathcal{G}}_{XY} \cup \mathcal{U}$
9                     Remove the edge $X - Y$ from $\mathcal{G}$;
10                     $\text{Sepset}(X, Y) \leftarrow (\mathcal{B}^{\mathcal{G}}_{XY} \cap \text{Adj}(\mathcal{G}, X)) \cup \mathcal{U}$;
11         $\bar{\mathcal{W}}_{XY} = \mathcal{W}^{\mathcal{G}}_{XY}$;
12     $m = m + 1$;
13 Orient the edges using the orientation rules in (Spirtes et al., 2000).

---

Table 5: Number of Nodes and Arcs

| DATASET | NODES | ARCS |
|---|---|---|
| EARTHQUAKE | 5 | 4 |
| CANCER | 5 | 4 |
| SURVEY | 6 | 6 |
| ASIA | 8 | 8 |
| CHILD | 20 | 25 |
| SACHS | 11 | 17 |
| ALARM | 37 | 46 |
| MILDEW | 35 | 46 |
| WIN95PTS | 76 | 112 |
| INSURANCE | 27 | 52 |
| WATER | 32 | 66 |
| HAILFINDER | 56 | 66 |
| HEPAR2 | 70 | 123 |
| MUNIN | 186 | 273 |
| ANDES | 223 | 338 |
| DIABETES | 413 | 602 |

Table 6: The number of CI tests for datasets with 100 samples

| DATASET | BPIT | OPT. BPIT | PIT | PC | PC-STABLE |
|---|---|---|---|---|---|
| EARTHQUAKE | **20** | **20** | **20** | 22 | 26 |
| CANCER | 10 | 10 | 10 | 10 | 10 |
| SURVEY | 15 | 15 | 15 | 15 | 15 |
| ASIA | **37** | **37** | **37** | 38 | 40 |
| CHILD | **268** | **268** | 275 | 323 | 389 |
| SACHS | **92** | **92** | 95 | 122 | 145 |
| ALARM | **1092** | **1092** | 1097 | 1199 | 1437 |
| MILDEW | 1006 | 1006 | **873** | 913 | 1003 |
| WIN95PTS | 3231 | 3231 | **3216** | 3360 | 3404 |
| INSURANCE | 642 | 642 | **619** | 661 | 764 |
| WATER | **559** | **559** | **559** | 576 | 591 |
| HAILFINDER | **2131** | **2131** | 2135 | 2205 | 2332 |
| HEPAR2 | **2514** | **2514** | 2521 | 2578 | 2606 |
| MUNIN | **40683** | **40683** | 40773 | 41643 | 40773 |
| ANDES | **25390** | **25390** | 25436 | 25778 | 26083 |
| DIABETES | 137400 | 121099 | **116671** | 119096 | 133959 |

Table 7: The number of CI tests for datasets with 1000 samples

| DATASET | BPIT | OPT. BPIT | PIT | PC | PC-STABLE |
|---|---|---|---|---|---|
| EARTHQUAKE | **39** | **39** | **39** | 55 | 60 |
| CANCER | **14** | **14** | **14** | 20 | 21 |
| SURVEY | **18** | **18** | **18** | 20 | 20 |
| ASIA | **96** | **96** | 100 | 103 | 115 |
| CHILD | **759** | **759** | 763 | 1156 | 1618 |
| SACHS | **471** | **471** | **471** | 771 | 966 |
| ALARM | 2042 | **2041** | 2073 | 2287 | 3353 |
| MILDEW | **1588** | **1588** | 1624 | 1675 | 2354 |
| WIN95PTS | **5654** | 5655 | 5736 | 5724 | 6914 |
| INSURANCE | **1571** | 1693 | **1571** | 1824 | 2448 |
| WATER | **702** | **702** | 707 | 770 | 873 |
| HAILFINDER | **13919** | 13990 | 14136 | 15245 | 20006 |
| HEPAR2 | **3041** | **3041** | 3196 | 4247 | 4783 |
| MUNIN | **143608** | 143648 | 144206 | 145403 | 144206 |
| ANDES | **29058** | 29119 | 32806 | 34253 | 38632 |
| DIABETES | 385937 | 381312 | **371386** | 371914 | 556224 |

Table 8: Runtime (seconds) for datasets with 100 samples

| DATASET | BPIT | OPT. BPIT | PIT | PC | PC-STABLE |
|---|---|---|---|---|---|
| EARTHQUAKE | **0.038** | 0.05 | 0.115 | 0.043 | 0.15 |
| CANCER | 0.012 | 0.02 | 0.012 | **0.12** | 0.13 |
| SURVEY | 0.019 | 0.03 | 0.019 | **0.018** | 0.019 |
| ASIA | 0.054 | 0.08 | **0.05** | 0.051 | 0.059 |
| CHILD | 0.85 | 1.06 | **0.53** | 0.7 | 1.04 |
| SACHS | 0.19 | 0.23 | **0.18** | 0.31 | 0.43 |
| ALARM | 4.7 | 5.7 | 2.6 | **2.5** | 3.7 |
| MILDEW | 19.3 | 20.5 | **5.2** | 5.4 | 4 |
| WIN95PTS | 33.4 | 46.8 | 10.7 | **4.3** | 4.6 |
| INSURANCE | 3 | 3.4 | **1.3** | 1.4 | 1.8 |
| WATER | 1.6 | 2.2 | 0.88 | **0.71** | 0.8 |
| HAILFINDER | 15.5 | 20.2 | 7.5 | **5.9** | 7.2 |
| HEPAR2 | 24.2 | 32.8 | 7.6 | **3.3** | 3.7 |
| MUNIN | 1686 | 2557 | 511 | **175** | 216 |
| ANDES | 2450 | 3901 | 487 | **29** | 30 |
| DIABETES | 34688 | 67043 | 6572 | **415** | 560 |

Table 9: Runtime (seconds) for datasets with 1000 samples

| DATASET | BPIT | OPT. BPIT | PIT | PC | PC-STABLE |
|---|---|---|---|---|---|
| EARTHQUAKE | **0.075** | 0.08 | 0.073 | 0.132 | 0.15 |
| CANCER | 0.018 | 0.023 | **0.017** | 0.34 | 0.38 |
| SURVEY | 0.024 | 0.034 | **0.23** | 0.029 | 0.03 |
| ASIA | 0.25 | 0.27 | **0.22** | 0.23 | 0.27 |
| CHILD | 4.12 | 4.5 | **2.64** | 10.7 | 14.2 |
| SACHS | 3.46 | 3.5 | **3.4** | 13.1 | 18.4 |
| ALARM | 12.2 | 13.6 | **8.3** | 9 | 17.2 |
| MILDEW | 47 | 48 | 30 | **27** | 38 |
| WIN95PTS | 54.7 | 70 | 22.5 | **12.2** | 16.1 |
| INSURANCE | 11.9 | 14.2 | **6.2** | 9.5 | 13.3 |
| WATER | 3 | 3.6 | 1.72 | **1.5** | 2 |
| HAILFINDER | 562 | 581 | **534** | 600 | 918 |
| HEPAR2 | 49 | 58 | **10.8** | 18.2 | 26.2 |
| MUNIN | 3323 | 4731 | 1766 | **1390** | 2801 |
| ANDES | 2697 | 5156 | 580 | **81** | 108 |
| DIABETES | 50617 | 91340 | **5262** | 9818 | 7577 |

Table 10: Structural Hamming Distance divided by the total number of true edges (%) for datasets with 100 samples

| DATASET | BPIT | OPT. BPIT | PIT | PC | PC-STABLE |
|---|---|---|---|---|---|
| EARTHQUAKE | 125 | 125 | 125 | **100** | 75 |
| CANCER | 100 | 100 | 100 | 100 | 100 |
| SURVEY | 100 | 100 | 100 | 100 | 100 |
| ASIA | **75** | **75** | 75 | 87.5 | 87.5 |
| CHILD | **56** | **56** | **56** | 80 | 84 |
| SACHS | **47** | **47** | 58.8 | 70.6 | 70.6 |
| ALARM | 80.4 | 80.4 | 78.2 | **63** | 76 |
| MILDEW | 128 | 128 | 121 | **87** | 91 |
| WIN95PTS | 96 | 96 | 94 | 92 | **90** |
| INSURANCE | 86.5 | 86.5 | 78.8 | **75** | 76.9 |
| WATER | 83.3 | 83.3 | 83.3 | 83.3 | 83.3 |
| HAILFINDER | 112 | 112 | 109 | **85** | 77 |
| HEPAR2 | 108 | 108 | 105 | 99 | **94** |
| MUNIN | 140 | 140 | 136 | **100** | 98.5 |
| ANDES | 95.5 | 95.5 | 90.5 | 81 | **79.6** |
| DIABETES | 242 | 242 | 99.3 | **73.8** | 74.8 |

Table 11: Structural Hamming Distance divided by the total number of true edges (%) for datasets with 1000 samples

| DATASET | BPIT | OPT. BPIT | PIT | PC | PC-STABLE |
|---------|------|-----------|-----|-----|-----------|
| EARTHQUAKE | **25** | **25** | **25** | 50 | 50 |
| CANCER | **50** | **50** | **50** | **50** | **50** |
| SURVEY | **50** | **50** | **50** | 67 | 67 |
| ASIA | **50** | **50** | **50** | **50** | **50** |
| CHILD | 32 | 32 | 20 | 32 | 32 |
| SACHS | **11.7** | **11.7** | **11.7** | 23.5 | 29 |
| ALARM | **26** | **26** | **26** | 30 | 35 |
| MILDEW | 63 | 63 | 54 | **43** | 48 |
| WIN95PTS | 72 | 73 | 71 | **63** | 66 |
| INSURANCE | 59.6 | 59.6 | **46.1** | 48 | 51.9 |
| WATER | **72.7** | **72.7** | **72.7** | **72.7** | 74 |
| HAILFINDER | 92 | 92 | 75 | **72** | **65** |
| HEPAR2 | **66** | **66** | 68 | 75 | 73 |
| MUNIN | 105 | 110 | 93 | **85** | **85** |
| ANDES | 63 | 68.6 | 44.7 | 43.8 | **43.5** |
| DIABETES | 133.5 | 133.5 | 67.6 | 67.1 | **63.3** |

Table 12: Runtime (second)

| DATASET | HILL-CLIMBING | | | TABU | | |
|---------|------|------|-------|------|------|------|
| | 100 | 1000 | 10000 | 100 | 1000 | 10000 |
| EARTHQUAKE | 0.07 | 0.08 | 0.09 | 0.08 | 0.07 | 0.09 |
| CANCER | 0.04 | 0.07 | 0.1 | 0.06 | 0.08 | 0.11 |
| SURVEY | 0.06 | 0.09 | 0.21 | 0.06 | 0.1 | 0.21 |
| ASIA | 0.16 | 0.22 | 0.33 | 0.17 | 0.24 | 0.35 |
| CHILD | 1.23 | 1.78 | 4.22 | 1.32 | 1.9 | 4.61 |
| SACHS | 0.32 | 0.43 | 0.8 | 0.34 | 0.46 | 0.83 |
| ALARM | 3.46 | 4.71 | 8.03 | 3.67 | 5.11 | 8.26 |
| MILDEW | 2.04 | 3.21 | 6.61 | 2.18 | 3.49 | 6.93 |
| WIN95PTS | 17.23 | 25.49 | 51.04 | 18.25 | 27.34 | 55.2 |
| INSURANCE | 2.03 | 2.81 | 5.72 | 2.2 | 3.01 | 6.3 |
| WATER | 2.42 | 2.8 | 5.01 | 2.61 | 3.06 | 5.31 |
| HAILFINDER | 14.96 | 22.28 | 53.21 | 15.92 | 23.91 | 56.46 |
| HEPAR2 | 13.14 | 14.97 | 30.75 | 14.64 | 15.76 | 33.01 |
| MUNIN | 5570 | 7266 | 15384 | 5877 | 7657 | 16213 |
| ANDES | 22059 | 25238 | 50506 | 22061 | 25566 | 72144 |
| DIABETES | 36791 | 99260 | 254561 | 38680 | 100303 | 251794 |

Table 13: Structural Hamming Distance divided by the total number of edges (Percent)

| DATASET | HILL-CLIMBING | | | TABU | | |
|---|---|---|---|---|---|---|
| | 100 | 1000 | 10000 | 100 | 1000 | 10000 |
| EARTHQUAKE | 75 | 50 | 50 | 75 | 50 | 50 |
| CANCER | 100 | 50 | 50 | 100 | 50 | 50 |
| SURVEY | 100 | 66.7 | 50 | 100 | 66.7 | 50 |
| ASIA | 62.5 | 50 | 50 | 62.5 | 50 | 50 |
| CHILD | 120 | 116 | 120 | 120 | 116 | 124 |
| SACHS | 76.4 | 52.9 | 47 | 76.4 | 52.9 | 47 |
| ALARM | 84.7 | 69.5 | 67.3 | 82.6 | 65.2 | 65.2 |
| MILDEW | 91.3 | 82.6 | 76 | 91.3 | 82.6 | 76 |
| WIN95PTS | 91 | 83 | 78.5 | 95.5 | 85.7 | 86.6 |
| INSURANCE | 109.6 | 105.7 | 98 | 109.6 | 105.7 | 98 |
| WATER | 87.8 | 90.9 | 81.8 | 87.8 | 87.8 | 80.3 |
| HAILFINDER | 168.1 | 189.3 | 225.7 | 168.1 | 189.3 | 225.7 |
| HEPAR2 | 105.6 | 85.3 | 6 66.6 | 106.5 | 85.3 | 68.2 |
| MUNIN | 105.1 | 1 93.4 | 87.1 | 106.9 | 91.2 | 86.4 |
| ANDES | 107.3 | 83.7 | 71.8 | 107.3 | 83.1 | 67.1 |
| DIABETES | 98 | 93.3 | 92.8 | 98 | 93.8 | 96 |

Table 14: Run time (second) using an oracle for the CI tests and 10000 samples

| DATASET | BPIT | OPT. BPIT | PC | PC-STABLE |
|---|---|---|---|---|
| EARTHQUAKE | **0.077** | 0.078 | 0.243 | 0.272 |
| CANCER | **0.084** | 0.085 | 0.265 | 0.297 |
| SURVEY | **0.289** | 0.29 | 0.796 | 0.961 |
| ASIA | **0.61** | **0.61** | 0.738 | 1.079 |
| CHILD | 44.35 | **33.48** | 106.40 | 131.1 |
| SACHS | 30.5 | 30.5 | **30.3** | 34.6 |
| ALARM | 71.5 | 69.6 | **65.5** | 127.8 |
| MILDEW | 1992 | 1878 | **1602** | 4384 |
| WIN95PTS | 472 | **450** | 752 | 1058 |
| INSURANCE | 1448 | 1322 | **1308** | 2489 |
| WATER | 5441 | 5374 | **5208** | 10172 |
| HAILFINDER | 550 | **462** | > 604800 | > 604800 |
| HEPAR2 | 7513 | **4630** | > 604800 | > 604800 |
| MUNIN | **27631** | 28763 | 78796 | 117688 |

Table 15: Runtime (second) using an oracle for the CI tests for 100 and 1000 samples

| DATASET | BPIT | | OPT. BPIT | | PC | | PC-STABLE | |
|---|---|---|---|---|---|---|---|---|
| | 100 | 1000 | 100 | 1000 | 100 | 1000 | 100 | 1000 |
| EARTHQUAKE | 0.055 | 0.068 | 0.068 | 0.067 | 0.19 | 0.3 | 0.21 | 0.22 |
| CANCER | 0.054 | 0.067 | 0.056 | 0.068 | 0.18 | 0.22 | 0.21 | 0.24 |
| SURVEY | 0.21 | 0.22 | 0.2 | 0.21 | 0.57 | 0.62 | 0.68 | 0.74 |
| ASIA | 0.4 | 0.43 | 0.4 | 0.43 | 0.48 | 0.51 | 0.69 | 0.75 |
| CHILD | 9.8 | 17 | 9.6 | 14.8 | 39 | 71 | 56 | 89 |
| SACHS | 14 | 22.6 | 14 | 23 | 14 | 23 | 17 | 26 |
| ALARM | 32 | 42 | 33 | 42 | 35 | 45 | 69 | 89 |
| MILDEW | 380 | 912 | 380 | 909 | 443 | 1021 | 1158 | 2794 |
| WIN95PTS | 177 | 263 | 208 | 259 | 208 | 403 | 307 | 557 |
| INSURANCE | 570 | 844 | 567 | 827 | 620 | 913 | 1289 | 1810 |
| WATER | 2410 | 3619 | 2398 | 3605 | 2385 | 3580 | 4478 | 6989 |
| HAILFINDER | 140 | 194 | 150 | 198 | 121752 | > 604800 | 232432 | > 604800 |
| HEPAR2 | 1454 | 2763 | 1518 | 2431 | > 604800 | > 604800 | > 604800 | > 604800 |
| MUNIN | 9775 | 94446 | 12197 | 54376 | 11160 | 27359 | 21558 | 44898 |