# OpenReview forum: "The Path-Driven Independence Testing (PIT) Algorithm"
_ICLR.cc/2025/Conference — Submitted to ICLR 2025_

### Official Review · Reviewer_LG3f · 2024-10-28

**Soundness:** 2
**Presentation:** 3
**Contribution:** 1
**Rating:** 3
**Confidence:** 4

**Summary:**

- The paper proposes to leverage the paths from the partially learned network to avoid conducting certain CI tests, leading to significant improvement in terms of time complexity. An important relevant work by Li et al. [1] is missing from the comparison in terms of accuracy and run time. A large body of relevant work is also missing from the paper. Overall, I think the paper presents a very incremental idea and the empirical comparison is not thorough given the existing literature.

**Reference:**

[1] Li, Honghao, et al. "Constraint-based causal structure learning with consistent separating sets." Advances in neural information processing systems 32 (2019).

**Strengths:**

- The proposed method aims to increase the efficiency of the PC algorithm by restricting the conditioning sets to neighbors that have paths to the target variable when verifying an edge.

**Weaknesses:**

- $\mathcal{I}(\mathcal{G})$ is not defined in the background.
- The phrase "activate the trail” is not defined in the background.
- The idea seems to be overlapping significantly with the work by Li et al. [1], which says to implement the CI tests for a pair X, Y with the conditioning set that contains only variables along the path between X and Y. Also, [1] is not cited in the paper.  Please see the first condition of defining a consistent set which is Definition 1 in [1]. The difference is that the proposed method tries to include more nodes to ensure the path is blocked whereas Li et al. exclude nodes that are common children from the conditioning set between the pair. In other words, Li et al.’s algorithm should subsume the proposed Algorithm 1, which is the main algorithm that shows significant improvement in reducing the number of CI tests.
- Although Algorithm 2 BPIT saves the number of CI tests, it also increases the conditioning set size which may lose statistical power more quickly in practice with a small sample size. This is evident by looking at the performance in Table 4 for the dataset DIABETES, which has 413 nodes and 602 arcs.
- There are plenty of algorithms that aim to reduce the number of CI tests by using a divide-and-conquer strategy for large-scale causal discovery [2,3,4,5] and none of them is cited and no discussion is presented to address the gap between the proposed method and these works. If the proposed method can be plugged into this line of work, then it is fair to pick a SOTA method and integrate the proposed strategy into it to show how much of a difference it can make.
- The experiment uses only Structual Hamming Distance as the only metric for evaluating model accuracy.


**Reference:**

[1] Li, Honghao, et al. "Constraint-based causal structure learning with consistent separating sets." Advances in neural information processing systems 32 (2019).

[2] Zhang, Hao, et al. "Learning causal structures based on divide and conquer." IEEE Transactions on Cybernetics 52.5 (2020): 3232-3243.

[3] Cai, Ruichu, Zhenjie Zhang, and Zhifeng Hao. "Sada: A general framework to support robust causation discovery." International conference on machine learning. PMLR, 2013.

[4] Zhang, Hao, et al. "Recursively learning causal structures using regression-based conditional independence test." Proceedings of the AAAI Conference on Artificial Intelligence. Vol. 33. No. 01. 2019.

[5] Zhang, Hao, et al. "Towards Effective Causal Partitioning by Edge Cutting of Adjoint Graph." IEEE Transactions on Pattern Analysis and Machine Intelligence (2024).

**Questions:**

- Can the authors address the similarities and differences between the proposed method and those proposed by Li et al. [1] and empirically compare the differences to show the benefits of the proposed method?
- Can the authors discuss the trade-off in more detail about using blockable paths and sample size? An analysis of how performance varies with sample size or conditioning set size across different datasets will be helpful.
- Can the proposed method be integrated into [2] and see how much improvement it can make on top of [2] in terms of the number of CI tests and SHD?
- Based on Table 10, it seems like PIT (Algorithm 1) will also struggle when the sample size is small and performs worse than PC when the graph gets larger, can the authors explain why?

**Reference:**

[1] Li, Honghao, et al. "Constraint-based causal structure learning with consistent separating sets." Advances in neural information processing systems 32 (2019).

[2] Zhang, Hao, et al. "Towards Effective Causal Partitioning by Edge Cutting of Adjoint Graph." IEEE Transactions on Pattern Analysis and Machine Intelligence (2024).

---

### Official Review · Reviewer_oSFj · 2024-10-30

**Soundness:** 3
**Presentation:** 3
**Contribution:** 2
**Rating:** 6
**Confidence:** 2

**Summary:**

This paper studies causal discovery for faithful DAG models, and the goal is to improve the classic PC algorithm in terms of runtime and number of conditional independence tests used throughout the procedure. It proposes an algorithm PIT, along with two variants, to improve PC algorithm based on the idea that when determining the link between any pair of nodes, it is unnessary to condition on all subsets of neighbours. Instead, PIT finds the ''essential parents'' and use them as conditioning sets. Some theoretical guarantee and complexity analysis are given for the proposed methods. Experiments are conducted to compare with PC algorithm.

**Strengths:**

- PC algorithm is the classic benchmark in faithful DAG learning. Many discussion are about how to reduce the runtime of PC.
- The proposed methods are shown to be sound and complete for the P-map. The number of CI tests is shown to be no worse than PC algorithm. Some example graphs are given in Table 1 to showcase strict improvement.
- Experiments on several datasets show competitive accuracy of PIT compared to PC but with overall faster runtime.

**Weaknesses:**

- The paper is dense in notation and new concepts in nature. A runnning example when introducing these concepts and algorithms will be helpful to illustrate the idea.
- The experiments and showcase of improvement leave some doubt. See questions.

**Questions:**

- In Table 1, it seems the proposed methods mainly show great improvement for hub graphs. Is there any other nontrivial class of graphs where the proposal can show significant improvement?
- Why the two variants of PIT are much slower than PC in Table 2? While they do not improve in terms of the number of CI test by much compared to PC.
- In Table 4, the two variants also show much worse accuracy compared to PC and PIT. Why bother introducing them?
- Any time complexity analysis apart from analysis for number of CI tests?
- I wonder if there is any attempt in literature on reducing the runtime or CI tests of PC algorithm?

---

### Official Review · Reviewer_yvJB · 2024-11-04

**Soundness:** 3
**Presentation:** 3
**Contribution:** 2
**Rating:** 5
**Confidence:** 4

**Summary:**

This work considers the Bayesian learning problem and develops a constraint-based method called PIT, which uses the partially learned network to reduce the number of conditional independence (CI) tests. The authors prove the soundness of their algorithm and evaluate its performance by comparing it to the PC algorithm.

**Strengths:**

The paper is well-written, addressing a relevant problem in Bayesian learning.
The proposed method is developed by exploring novel and interesting properties of Bayesian networks.
The empirical results are promising.

**Weaknesses:**

Although the proposed method is theoretically sound and Table 1 presents the number of CI tests required by PIT for certain network structures, there is no guarantee for general structures. As a result, it is unclear whether PIT is faster than PC or other constraint-based methods. An analysis of the worst-case number of CI tests is missing. For example, in "A Recursive Markov Boundary-Based Approach to Causal Structure Learning" by Mokhtarian et al., the authors propose a recursive method, analyze the worst-case number of CI tests, compare it with other methods (not only PC), and introduce a lower bound. How does PIT compare to these methods?

The empirical results are limited, as the authors only compare PIT with PC and omit comparisons with other relevant methods such as Marvel, GS, and CS.

The complexity of constraint-based methods depends not only on the number of CI tests but also on the size of the conditioning set. How large could the conditioning sets be in the PIT algorithm?

This method, like many other constraint-based approaches, requires the causal sufficiency assumption, which often does not hold in practice. However, some relevant algorithms can be extended to handle cases with latent confounders. Based on PIT's logic, it is unclear whether the notion of path-driven independence tests can be extended to networks with latent confounders. Do the authors have any insights on how this method might be generalized?

**Questions:**

Please see the above comments.

---

### Official Review · Reviewer_mE6H · 2024-11-04

**Soundness:** 4
**Presentation:** 4
**Contribution:** 2
**Rating:** 3
**Confidence:** 3

**Summary:**

The manuscript describes a new structure learning algorithm for Bayesian networks called Path-Driven Independence Testing (PIT). The proposed method is an improvement over the traditional  PC algorithm, aiming to reduce computational cost by minimizing the amount of conditional independence (CI) tests. The paper introduces concepts such as “covering paths” and “blindly blockable paths” between CI variables to minimize the number of tests while maintaining accuracy.

**Strengths:**

* Well-written paper with clear presentation and pace, including helpful examples and theoretical background.
* The central insight of the proposed method, described in Section 3.1, can lead to a significantly reduced amount of independent tests and, thus, time savings, as shown in Section 4.

**Weaknesses:**

* My main concern with the manuscript is the relevance of its main contribution when compared to existing methods in the literature. The paper could benefit from an in-depth related work analysis on methods that had proposed time savings when testing independence, such as [1] or [2] and others. The relevance of the method proposed here could be better demonstrated when concepts in the manuscripts are compared to existing concepts. For instance, the idea of Covering Path (Definition 1) and the result in Lemma 3 would fare in comparison with other algorithms' active/relevant nodes.
* Although experimental results are encouraging, the paper is unclear if the wins are consistent or significant in structural learning timings. Table 2, for instance, shows favorable results, although for larger datasets, the gains could be significant in some networks while very close to others. The manuscript could benefit from a thorough analysis of the results so that algorithm users might manage expectations when using it in specific circumstances or applications.

[1] Shachter, Ross D. "Bayes-ball: The rational pastime (for determining irrelevance and requisite information in belief networks and influence diagrams)."

[2]  Butz, Cory J., André E. dos Santos, and Jhonatan S. Oliveira. "Relevant path separation: A faster method for testing independencies in Bayesian networks."

**Questions:**

* Minor: I couldn’t find the definition of N in N_{X_5} in Example 5

---

### Meta-Review · Area_Chair_RSrT · 2024-12-20

**Metareview:**

The reviewers found novelty to be limited. Importantly, some of them also identified important related work with significant overlap that was missed by the authors. The advantage in terms of CI test was also unclear according to some reviewers.

The authors did not submit a rebuttal, effectively admitting that the paper needs a significant revision.

**Additional Comments On Reviewer Discussion:**

The authors didn’t submit a rebuttal.

---

### Decision · Program_Chairs · 2025-01-22

Reject